# The Mechanism of Sodium Sulfate Coupled with Anaerobic Methane Oxidation Mitigating Methane Production in Beef Cattle

**DOI:** 10.3390/microorganisms12091825

**Published:** 2024-09-03

**Authors:** Xiaowen Zhu, Zhiyu Zhou, Yang Cheng, Ziqi Deng, Hao Wu, Luiz Gustavo Nussio, Zhenming Zhou, Qingxiang Meng

**Affiliations:** 1State Key Laboratory of Animal Nutrition and Feeding, College of Animal Science and Technology, China Agricultural University, Beijing 100193, China; zxw1542705411@163.com (X.Z.);; 2Forage Quality and Conservation Lab, Department of Animal Sciences, Luiz de Queiroz College of Agriculture, University of Sao Paulo, Sao Paulo 05508-220, Brazil

**Keywords:** rumen, methane emission reduction, anaerobic oxidation of methane

## Abstract

The aim of this experiment is to explore the effect of sodium sulfate (Na_2_SO_4_) on methane reduction in the rumen, and its impact on anaerobic methane-oxidizing archaea (ANME). Using mixed rumen fluid from four Angus cattle fistulas, this study conducted an *in vitro* fermentation. Adding Na_2_SO_4_ to the fermentation substrate resulted in sulfur concentrations in the substrate of 0.4%, 0.6%, 0.8%, 1.0%, 1.2%, 1.4%, 1.6%, 1.8%, 2.0%, 2.2%, and 2.4%. The gas production rate and methane yield were measured using an *in vitro* gas production method. Subsequently, the fermentation fluid was collected to determine the fermentation parameters. The presence of ANME in the fermentation broth, as well as the relationship between the number of bacteria, archaea, sulfate reducing bacteria (SRB), ANME, and the amount of Na_2_SO_4_ added to the substrate, were measured using qPCR. The results showed that: (1) the addition of Na_2_SO_4_ could significantly reduce CH_4_ production and was negatively correlated with CO_2_ production; (2) ANME-1 and ANME-2c did exist in the fermentation broth; (3) the total number of archaea, SRB, ANME-1, and ANME-2c increased with the elevation of Na_2_SO_4_. The above results indicated that Na_2_SO_4_ could mitigate methane production via sulfate-dependent anaerobic methane oxidation (S-DAMO) in the rumen. In the future management of beef cattle, including sodium sulfate in their diet can stimulate S-DAMO activity, thereby promoting a reduction in methane emissions.

## 1. Introduction

Methane (CH_4_) has a very high global warming potential, which is 28 times that of carbon dioxide (CO_2_). Methane emissions from ruminants account for approximately 3% to 5% of total global greenhouse gas emissions [1]. They constitute a significant 80% of the methane emissions produced by the livestock industry [2], and they represent 2% to 12% of the energy loss in ruminants. With the continuous improvement of people’s living standards and the increasing demand for meat and milk, gastrointestinal CH_4_ emissions from ruminants are expected to increase significantly. Therefore, mitigating CH_4_ emissions from ruminants is crucial for enhancing feed efficiency and achieving the long-term goal of carbon neutrality. Sulfur is a crucial trace element required for the growth and development of beef cattle. Studies have shown that incorporating Na_2_SO_4_ into beef cattle’s diets can significantly diminish CH_4_ production. Sandeep, U, et al. [3] found that when Na_2_SO_4_ was exogenously added to the diet of goats, methane production in the rumen could be significantly reduced when the substrate sulfur concentration reached 0.76% (on a dry matter basis). Wu et al. [4] found that the *in vitro* addition of Na_2_SO_4_ at a sulfur concentration of 1.39% of DM in the substrate significantly reduced CH_4_ production. Wu et al. [5] demonstrated that adding Na_2_SO_4_ to beef cattle’s diets to adjust the substrate sulfur concentration to 1.0% can also reduce CH_4_ production, and it can increase the permeability of the rumen epithelium, while having no significant impact on the rumen microbial community. Studies have shown that adding Na_2_SO_4_ (with a sulfur concentration of 0.225% DM) to the ruminant diet enhances the activity of fiber-digesting bacteria and rumen fungi [6]. Feeding a total mixed ration (TMR) soaked in sulfate can increase the degradation rate of cellulose [7]. Adding Na_2_SO_4_ with a substrate sulfur concentration of 0.64% can promote the growth and reproduction of rumen micro-organisms [8], and supplementing the diet with Na_2_SO_4_ (with a sulfur concentration of 0.19% DM) does not have any adverse effects on rumen fermentation. However, the effect of Na_2_SO_4_ on the reduction of CH_4_ emissions in the rumen is generally accepted as an outcome caused by SO_4_^2−^ competing with CO_2_ for hydrogen atoms to form HS^−^ [9]. In the natural environment, a type of pathway called sulfate-dependent anaerobic methane oxidation (S-DAMO) has a capacity to reduce methane by up to 90% [10]. This mechanism operates chiefly via the reverse methanogenesis pathway, where the transfer of electrons converts CH_4_ into CO_2_, thereby achieving this effect [11]. The final product of the reaction is HS^−^, which is consistent with the rumen environment [12]. Its occurrence conditions are strictly anaerobic, and the pH is similar to that of the rumen environment [13]. Hence, we hypothesize that the rumen harbors the conducive conditions for the occurrence of S-DAMO, given its three defining features: decreased methane production, the presence of sulfate-reducing bacteria (SRB), and the occurrence of anaerobic methanotrophic archaea (ANME) [14]. To this end, we plan to employ a dual approach of *in vitro* gas production experiments coupled with qPCR to ascertain whether sulfate supplementation can diminish methane emission in the rumen, and to identify the presence of ANME and SRB populations within the rumen environment, in order to provide new ideas and methods for methane mitigation in ruminants and data support for the study of S-DAMO in the rumen.

## 2. Materials and Methods

### 2.1. Experimental Design

A completely random single-factor design was employed to study the effects of sulfur on *in vitro* fermentation parameters, methane emissions, and rumen microbiota. Na_2_SO_4_ was used to adjust the sulfur content of the substrate, whose basic level was set at 0.4% (basal sulfur content of the substrate dry matter), 0.6% (using sodium sulfate as exogenous sulfur source), 0.8%, 1.0%, 1.2%, 1.4%, 1.6%, 1.8%, 2.0%, 2.2%, and 2.4%. In this experiment, the standard diet containing 0.4% sulfur was used as the control group, with no additives, reflecting the typical sulfur concentration found in TMR for beef cattle in conventional farming.

### 2.2. Management of Experimental Animal

Four castrated Angus cattle, each with a permanent rumen fistula and weighing approximately 500 kg, were selected as experimental animals. They were fed daily at 8:00 am and 4:00 pm, a 5% allowance for leftover feed was ensured, and animals had free access to water throughout the experiment. All animal experiments were approved by the Animal Welfare and Ethical Committee of the China Agricultural University (AW81404202-1-3). The feeding standards were based on the ‘Beef Cattle Nutrition Requirements’ of the National Academy of Science, Engineering, and Medicine (NASEM). The composition and nutritional levels of the diet are presented in Table 1.

### 2.3. Experiment Method

Fresh rumen fluid was collected from four Angus steers with permanent rumen fistulas, and the fluid was filtered through four layers of gauze before the morning feed. Artificial saliva was prepared according to Menken et al. [15]: 400 mL distilled water + 0.1 mL solution A + 200 mL solution B + 200 mL solution C + 1 mL Tianqing oxidation-reduction indicator + 40 mL reducing solution. Under a continuous flux of CO_2_, the rumen fluid and artificial saliva were mixed in a 1:2 (*v*/*v*) ratio to prepare the *in vitro* fermentation inoculum.

A total of 200 mg of feed sample (dry matter basis) was weighed and placed at the bottom of a 100-mL glass syringe (Deli Electric Power Equipment, Shijiazhuang, Hebei, China). Different concentrations of Na_2_SO_4_ solution were added to the fermentation substrate to adjust the S content. There were 11 treatments, each treatment containing 6 syringes. A total of 30 mL of inoculum was added to each syringe and incubated at 39 °C in an automatic shaker for 48 h (Jie Cheng Experimental Apparatus, Shanghai, China). During this period, 3 syringes were taken from each group at 24 h for the determination of pH, VFA, NH_3_-N, and gas components. At the end of the 48-h experiment, 3 syringes were taken out to measure pH, VFA, NH_3_-N, and gas components. The volume of the cumulative gas production was recorded manually at time points of 0, 2, 4, 6, 8, 10, 12, 16, 20, 24, 30, 36, 42, and 48 h.

### 2.4. Measurement Indicators and Methods

Volatile fatty acids (VFA) and gas components (CH_4_, CO_2_) were determined using gas chromatography. Ammonia nitrogen (NH_3_-N) was measured using the phenol sodium hypochlorite colorimetric method. pH was measured using a portable pH meter. The H_2_S content was determined using GASTEC fast gas detection tubes (Japan). Dry matter (DM) was analyzed using the AOAC method, and neutral detergent fiber (NDF) was determined using the Van Soest detergent fiber method. DNA was extraced using a DNA extraction kit (Tiangen Biotech Co., Beijing, China). The DNA quality was assessed by agarose gel (1%) electrophoresis, and DNA concentrations were measured using a NanoDrop 2000 Spectrophotometer (Thermo Fisher Scientific, Waltham, MA, USA) and stored at −20 °C. The count of total bacterial, archaea, sulfate-reducing bacteria, and ANME were quantified using an ABI 7300 Prism real-time PCR (ABI, Foster City, CA, USA) with SYBR Green PCR RealMaster Mix (Cwbio, Beijing, China) staining method. The reaction system is shown in Table 2. The specific reaction procedure was as follows: pre-denaturation at 95 °C for 10 min; denaturation at 95 °C for 10 s; annealing at 56–64 °C for 30 s; extension at 72 °C for 32 s, repeating steps 2–4 for 35–40 cycles; dissolution curve analysis: 95 °C for 15 s; 60 °C for 1 min; 95 °C for 15 s; 60 °C for 15 s.

### 2.5. Statistical Analysis

The dynamic parameters of gas production were calculated according to the formula of Menke et al. [15], with Y = B × (1 − e^−ct^) (the formula represents the change in gas production over time; B: the asymptotic gas production; c: the rate of gas production). All data were preliminarily calculated and organized using Office 365 Excel (version number 1808). Employing SPSS 27.0’s General Linear Model (GLM), a one-way ANOVA was performed, followed by subsequent multiple comparison tests to elucidate significant differences among groups. The orthogonal polynomial comparison method was used for first and second linear fitting, and the correlation analysis was conducted using the bivariate Pearson test, with a significance level of *p* < 0.05. Images were created using GraphPad Prism 10.

## 3. Results

### 3.1. Effect of Different Sulfur Concentrations on Gas Production and Fermentation Parameters In Vitro

As shown in Figure 1, during the *in vitro* fermentation the 11 different treatment groups exhibited a similar trend in gas production. There was no lag at the onset, and the rates of gas production were rapid during the first 12 h. But with the extension of the time, the curve gradually flattened and entered the plateau period at 48 h.

At each time point, the group with a concentration of 0.4% (control group) showed the highest gas production, while the group with a concentration of 2.2% had the lowest gas production. As indicated in Table 3, the total gas production in all groups decreased linearly with the increase in sulfur content in the substrates. The theoretical maximum gas production (B value) displayed a linear decrease trend, whereas the rate of gas production (c value) showed a quadratic linear increase. The findings of the study reveal that incorporating Na_2_SO_4_ into the mixture leads to a decrease in total gas production.

### 3.2. Effect of Different Sulfur Concentrations on the Gas Composition of Rumen Fermentation In Vitro

As shown in Figure 2 and Table 4, with the increase in sulfur dose, the concentration of H_2_S gradually increased and stabilized at 2.0%. Compared to the control group (sulfur concentration of 0.4%), the methane production in the first four groups was significantly reduced (sulfur concentration of 0.6–1.2%), and the methane production in the groups with a sulfur concentration of 1.4–2.0% was also lower than that of the control group. However, the methane production in the last two groups (sulfur concentration of 2.2–2.4%) increased. Figure 3 shows that adding sulfur increased the proportion of CO_2_ in the gas composition when compared to the control group. Once the sulfur concentration surpassed 2.2%, the CO_2_ proportion modestly declined. Table 5 shows the significant negative correlation between the sulfur concentration in the fermentation substrate and the proportion of CH_4_, as well as the production of CH_4_ and CO_2_.

### 3.3. Effect of Different Sulfur Concentrations on Rumen Fermentation Parameters In Vitro

Table 6 presented the fermentation parameters at 24 and 48 h, respectively. Despite the increasing sulfur content in the substrate, pH levels remained within the normal range, and there was no significant difference in NH_3_-N among the groups *(p >* 0.05). Similarly, the proportion of TVFA and various VFA did not exhibit significant changes *(p >* 0.05). Moreover, the type of *in vitro* fermentation remained unchanged (*P_24 h_* = 0.69, *P_48 h_* = 0.52).

### 3.4. PCR Validation of Methane Anaerobic Oxidizing Archaea and Optimization of PCR 

ConditionThe extracted total DNA was evaluated for quality using 1% agarose gel electrophoresis, as shown in Figure 4. The results revealed clear bands without extraneous bands, indicating that the DNA is suitable for subsequent amplification with specific primers and qPCR amplification.

Five pairs of primers were selected for PCR amplification and the optimization of amplification conditions. As shown in Figure 5, all five pairs of primers successfully amplified the corresponding DNA bands, which were subjected to gel extraction for subsequent qPCR standard preparation. The reaction conditions were optimized (Table 7), and the most suitable annealing temperature was determined for each pair of primers: the annealing temperature for Archaea806F/8958R and Bacteria BAC27F/EUB338R is 55 °C; ANME-1 1337F/1724R is 54.4 °C; ANME-2c 468F/736R is 56.3 °C; SRB F/R is 60 °C.

### 3.5. Effect of Different Sulfur Concentrations on Archaea, Bacteria, SRB and ANME

qPCR was employed to ascertain the abundance of archaea, total bacteria, SRB, and ANME within the fermentation. As shown in Table 8, there were significant differences in the number of Archaea, SRB, and ANME among different substrate sulfur concentrations, and an increase was observed with the increase in sulfur concentration, while the total number of bacteria did not show a significant change with the increase in sulfur concentration.

## 4. Discussion

### 4.1. Effect of Adding Na_2_SO_4_ on Gas Production Parameters and Gas Composition

The cumulative gas production and the gas parameters from the *in vitro* gas production assay are indicative of the ruminal fermentation capacity to a certain extent. Sulfur is a pivotal element for the proliferation of ruminal micro-organisms and the normalcy of cellular metabolism. The National Research Council (NRC), in its publication “Nutrient Requirements for Beef Cattle”, recommends a dietary sulfur concentration of 0.15% to support the growth of beef cattle. Previous studies have indicated that adding Na_2_SO_4_ to the diet of ruminants can reduce methane emissions, which has a positive effect on mitigating the greenhouse effect. Patra, A.K.’s [22] and Wu et al.‘s [5] *in vitro* trials have proven that the addition of Na_2_SO_4_ can reduce the total gas production and CH_4_ emissions. Hunerberg et al.’s [23] and Wu et al.‘s [4] in vivo trials also showed that adding Na_2_SO_4_ can reduce the methane emissions from ruminants. The results of this experiment are consistent with previous findings, showing that the total gas production decreases with an increase in Na_2_SO_4_ dosage. Furthermore, compared to the control group (0.4%), the methane production in the 0.6–1.2% group was significantly reduced, and the methane production in the 1.4–2.0% group was also lower than that in the control group. The correlation analysis also showed that methane production was significantly negatively correlated with the addition of sodium sulfate, which is consistent with the results of PATTAYA [24] and ABDL-RAHMA et al. [25]. These results all indicate that the addition of an appropriate amount of sodium sulfate to the diet of ruminants can, indeed, reduce methane emissions. However, the methane production in the 2.2% and 2.4% groups increased, while the carbon dioxide production decreased. This phenomenon may be postulated as arising from the enzymatic threshold being reached within the reaction system. As the enzymes mediating CO_2_ conversion to CH_4_ in the rumen are all bidirectional enzymes [26,27], enzyme inhibition reactions occurred [28]. When a certain amount of Na_2_SO_4_ was added to the substrate, this type of enzyme could promote the reverse production of methane by converting CH_4_ to CO_2_, resulting in a decrease in CH_4_ production. At substrate S concentrations surpassing 2.2%, these enzymes appear to enhance CH_4_ synthesis. Correlation analyses further substantiate a significant inverse relationship between CH_4_ and CO_2_ levels. The essence of AMO is to achieve the conversion of CH_4_ to CO_2_ [29], and its functional enzymes are also the same as those in the rumen [30,31]. Consequently, it can be inferred that there is a process of CH_4_ conversion to CO_2_ in the rumen, and S-DAMO exists in the rumen.

In addition to methane production, attention should also be paid to the content of H_2_S. An excessive intake of Na_2_SO_4_ can be metabolized into H_2_S by sulfate-reducing bacteria (SRB), which increases the risk of polioencephalomalacia (PEM) in livestock [32]. Gould et al. [33] demonstrated that the administration of 1.8% Na_2_SO_4_ in the feed of Holstein steers weighing 120–160 kg manifested symptoms of PEM. Conversely, other studies have reported that high-sulfur diets (1.72%) did not induce PEM in calves and sheep [34]. This study showed that the concentration of H_2_S increased correspondingly with the incremental addition of Na_2_SO_4_, reaching a stable level after the 2.0% concentration threshold was surpassed. However, this study is an *in vitro* experiment, and the tolerability of such H_2_S concentrations in vivo, relative to the weight and dietary composition of beef cattle, remains to be ascertained.

### 4.2. Effect of Adding Na_2_SO_4_ on Fermentation Parameters In Vitro

The ruminal fermentation parameters serve as direct indicators of the microbial activity and metabolism of fermentation products. The stability of the ruminal internal environment is paramount for fermentative processes. The appropriate pH levels and NH_3_-N concentrations are crucial for maintaining the stable ruminal internal environment, and ensuring the normal growth and reproduction of rumen micro-organisms. In this study, the pH levels across all groups fell within the normal range, showing no significant differences. Similarly, the NH_3_-N concentrations were within the effective range for microbial protein synthesis (5.0–30 mg/dL) [35]. Regarding volatile fatty acids, including acetate, propionate, butyrate, valerate, isobutyrate, isovalerate, and the acetate to propionate ratio, there were no significant differences observed with the increasing addition of Na_2_SO_4_, in agreement with the findings of Wu et al. [5]. However, in vivo studies have indicated a marked elevation in butyrate levels when the sulfur concentration reaches 1.0%. This discrepancy may be attributed to the limitations of *in vitro* experiments to fully replicate the complex ruminal environment, necessitating further in vivo experiments for validation.

### 4.3. Effect of Adding Na_2_SO_4_ on S-DAMO in the Rumen

Anaerobic methane oxidation (AMO) is a prevalent and promising pathway for methane mitigation within natural ecosystems [6], with sulfate-dependent anaerobic methane oxidation (S-DAMO) being a significant one. The ruminal environment, characterized by strict anaerobiosis and a pH ranging from 6.0 to 7.0, produces HS^−^ as a final product, which is highly consistent with the occurrence conditions of S-DAMO [36]. Thermodynamically, the reaction CH_4_ + SO_4_^2−^→HCO_3_^−^ +HS^−^ +H_2_O is more favorable than the process of SO_4_^2−^ + 4H_2_^+^H^+^→HS^−^ + 4H_2_O [37], suggesting the potential existence of S-DAMO in the rumen; ANME of S-DAMO is essential, and it has been confirmed that the S-DAMO process is mediated by micro-organisms [38] and is closely related to SRB. Typically, ANME and SRB engage in a symbiotic relationship, with ANME-1 associating loosely with SRB, whereas ANME-2 displays a tighter association [39,40]. Studies have shown that supplementing beef cattle diets with Na_2_SO_4_ can increase the counts of total archaea and SRB [32], indicating the possible presence of S-DAMO in the rumen. Therefore, this study focused on detecting total bacteria, total archaea, SRB, ANME-1, and ANME-2c in the fermentation broth. PCR amplification, qPCR, and gel electrophoresis revealed corresponding bands, providing preliminary evidence of ANME in the fermentation broth. The qPCR results indicated a significant increase in the total numbers of archaea, SRB, ANME-1, and ANME-2c, with higher Na_2_SO_4_ levels in the substrate. These findings support the presence of ANME in the rumen, influenced by sulfur concentration. The data indicate the presence of S-DAMO within the rumen, and, to gain further insights, high-throughput sequencing can be utilized to explore potential elevations in its expression levels.

## 5. Conclusions

In this study, a total of eleven addition levels of sodium sulfate were set to achieve substrate sulfur concentrations of 0.4%, 0.6%, 0.8%, 1.0%, 1.2%, 1.4%, 1.6%, 1.8%, 2.0%, 2.2%, and 2.4%. This was demonstrated through *in vitro* gas production experiments showing that adding a certain dose of Na_2_SO_4_ to the diet of ruminants can effectively reduce methane emissions. However, excessive addition that causes the sulfur concentration in the diet to exceed 2.2% DM will result in the loss of methane reduction effectiveness. Additionally, the addition of sodium sulfate had no significant effect on the fermentation parameters. Using the dutilization of PCR and qPCR techniques to affirm the presence of ANME in the rumen, it was noted that their abundance increased with the addition of Na_2_SO_4_, indicating that the S-DAMO process exists in the rumen However, the precise mechanistic role of ANME in the rumen, and the specific tolerance of Na_2_SO_4_ at different growth stages of beef cattle, still require further in vivo experiments to verify.

## Figures and Tables

**Figure 1 microorganisms-12-01825-f001:**
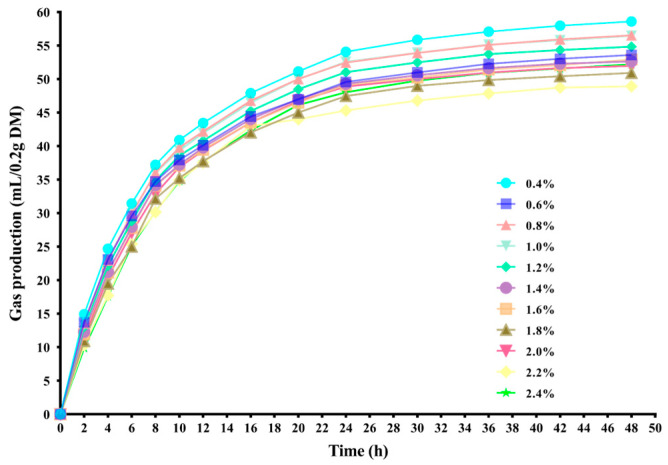
Dynamic changes of gas production of *in vitro* ruminal fermentation in different sulfur concentrations.

**Figure 2 microorganisms-12-01825-f002:**
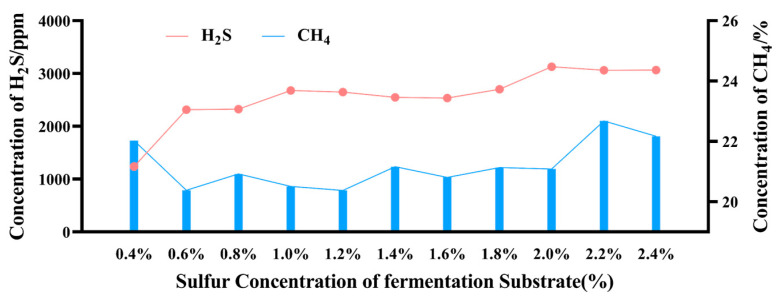
Effects of different sulfur levels on gas components of concentration of CH_4_ and H_2_S *in vitro* fermentation.

**Figure 3 microorganisms-12-01825-f003:**
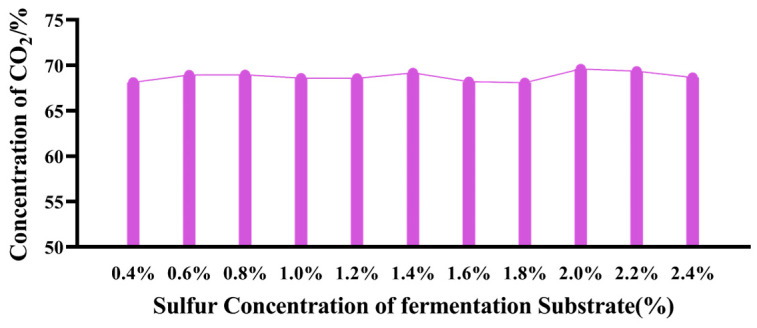
Effects of different sulfur levels on gas components of concentration of CO_2_ *in vitro* fermentation.

**Figure 4 microorganisms-12-01825-f004:**
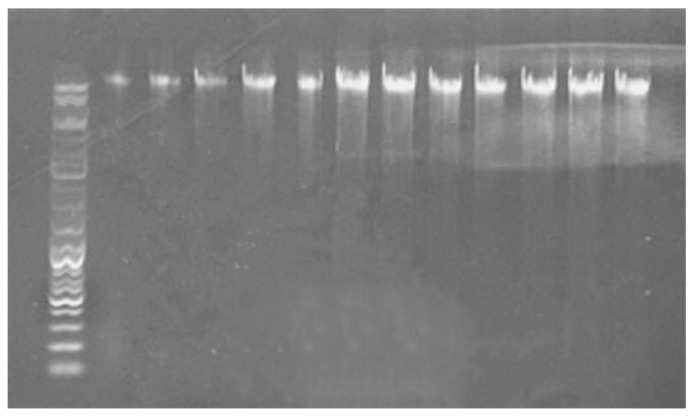
Quality examination of metagenomic DNA using agarose gel (1%).

**Figure 5 microorganisms-12-01825-f005:**
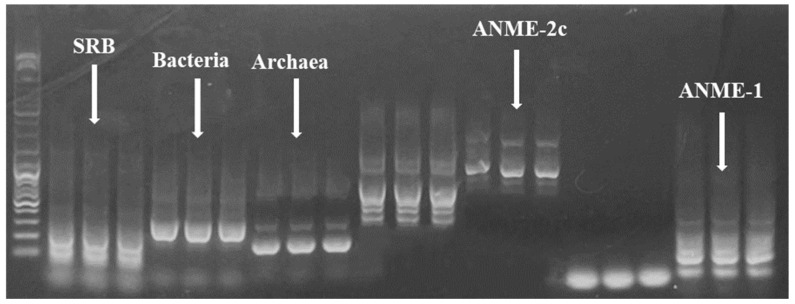
Quality examination of PCR amplification products sing agarose gel (1%).

**Table 1 microorganisms-12-01825-t001:** Composition and nutritional level of the diet (DM basis).

Items	Contents
Ingredients (g/kg DM)	
Whole plant corn silage	310.00
Corn stalker	130.00
Ground corn	320.00
Jujube powder	60.00
Soybean meal	80.00
Palm meal	60.00
Premix	17.00
Sodium bicarbonate	8.00
Salt	10.00
Total	1000.00
Nutritional composition	
ME (MJ/kg)	10.72
CP (%DM)	9.88
NDF (%DM)	49.02
ADF (%DM)	22.92
Ca (%DM)	0.60
P (%DM)	0.27
S (%DM)	0.40
Na_2_SO_4_ (%DM)	0.00

Note: each kilogram of premix contained the following: vitamin A acetate 150–450 thousand IU, vitamin D 340–120 thousand IU, Mn 1000–3000 mg, Fe 1000–5000 mg, Zn 1500–3700 mg, Cu 250–750 mg, Ca 10–25%, total *p* ≥ 0.3%, NaCl 15–30%, H_2_O ≤ 12%; metabolic energy is the calculated value, while other components are the measured values. ME: metabolic energy. CP: crude protein. NDF: neutral detergent fiber. ADF: acid detergent fiber. Ca: calcium. P: phosphorus. S: sulfur. Na_2_SO_4_: sodium sulfate.

**Table 2 microorganisms-12-01825-t002:** qPCR reaction mixture.

Items	Reaction System 50 μL
2 × Ultra SYBR Mixture (High ROX)	25
Primer-Forward (μM)	1
Primer-Reverse (μM)	1
Template DNA	2
ddH_2_O	up to 50

**Table 3 microorganisms-12-01825-t003:** Effects of different sulfur levels on gas production parameters *in vitro*.

Items	Treatments											SEM	*p*-Value	*p*-Value	
	0.4%	0.6%	0.8%	1.0%	1.2%	1.4%	1.6%	1.8%	2.0%	2.2%	2.4%			L	Q
Gas production (mL/0.2 g DM)															
24 h	54.07	49.57	52.47	52.57	51.03	49.30	49.13	47.47	48.88	44.82	48.03	0.79	0.20	<0.01	0.91
48 h	58.60	53.60	56.53	56.47	54.83	52.63	52.93	50.93	51.97	48.92	52.20	0.84	0.21	<0.01	0.82
Gas production parameters															
B (mL/0.2 g DM) ^1^	58.10	52.94	55.97	55.98	54.47	52.26	52.12	50.70	51.69	48.41	52.16	0.72	0.24	<0.01	0.57
C (h^−1^) ^2^	0.11	0.11	0.11	0.11	0.11	0.12	0.12	0.11	0.12	0.13	0.11	<0.01	0.11	0.48	0.02

^1^: the asymptotic gas production (mL/0.2 g DM); ^2^: the rate of gas production (h^−1^).

**Table 4 microorganisms-12-01825-t004:** Effects of different sulfur levels on gas components of concentration of CH_4_ and H_2_S *in vitro* fermentation.

Items	Treatments											SEM	*p*-Value
	0.4%	0.6%	0.8%	1.0%	1.2%	1.4%	1.6%	1.8%	2.0%	2.2%	2.4%		
CH_4_ (%)	22.03 ^bcd^	20.38 ^a^	21.92 ^bcd^	20.51 ^a^	20.38 ^a^	21.16 ^abc^	20.81 ^ab^	21.13 ^abc^	21.08 ^abc^	22.68 ^d^	22.17 ^cd^	0.24	<0.01
H_2_S (ppm)	1237.83 ^a^	2314.14 ^b^	2326.56 ^b^	2678.64 ^c^	2646.91 ^c^	2549.34 ^c^	2534.57 ^c^	2701.84 ^c^	3127.49 ^d^	3062.67 ^d^	3064.56 ^d^	157.66	<0.01

Note: The letters a–d denote significant mean differences, as established by Tukey’s honest significant difference (HSD) post-hoc analysis, with a significance level of *p* < 0.05.

**Table 5 microorganisms-12-01825-t005:** Correlation analysis of different sulfur levels and the proportion of CH_4_ and CO_2_.

Items	Sulfur Concentration (%)	CH_4_ (%)	CO_2_ (%)
Sulfur concentration (%)	1		
CH_4_ (%)	−0.628 **	1	
CO_2_ (%)	0.346	−0.478 **	1

** mean extremely significant correlation at *p* < 0.01.

**Table 6 microorganisms-12-01825-t006:** Effects of different sulfur levels on rumen fermentation *in vitro*.

Items	Treatments											SEM	*p*-Value	*p*-Value	
	0.4%	0.6%	0.8%	1.0%	1.2%	1.4%	1.6%	1.8%	2.0%	2.2%	2.4%			L	Q
24 h Fermentation parameter															
pH	6.61	6.61	6.60	6.56	6.58	6.58	6.59	6.57	6.61	6.64	6.58	<0.01	0.63	0.19	0.66
NH_3_-N ^1^ (mg/100 mL)	13.09	14.94	12.17	14.80	11.99	13.09	12.72	11.99	14.80	12.80	15.12	0.38	0.46	0.40	0.08
TVFA ^2^ (mmol/L)	71.99	72.55	61.84	66.72	66.60	70.20	65.81	67.64	80.09	58.34	58.94	1.92	0.80	0.45	0.71
Acetate (%)	65.56	63.99	63.96	64.40	64.63	64.78	65.93	64.91	66.85	64.68	63.60	0.29	0.73	0.73	0.57
Propionate (%)	17.52	17.36	17.28	17.04	17.69	17.46	17.12	17.47	17.69	16.89	17.52	0.08	0.06	0.68	0.59
Isobutyrate (%)	1.64	1.76	1.76	1.78	1.67	1.70	1.65	1.65	1.55	1.74	1.82	0.02	0.82	0.86	0.52
Butyrate (%)	11.13	11.96	12.06	11.59	11.59	11.70	10.91	11.47	10.47	11.68	11.99	0.15	0.66	0.64	0.64
Isovalerate (%)	2.91	3.49	3.50	3.66	3.12	3.09	3.08	3.20	2.70	3.54	3.57	0.09	0.45	0.93	0.60
Valerate (%)	1.23	1.45	1.43	1.52	1.29	1.27	1.31	1.29	1.04	1.46	1.50	0.04	0.46	0.82	0.53
A/P^3^	3.74	3.69	3.70	3.78	3.65	3.71	3.86	3.72	3.84	3.83	3.63	0.02	0.69	0.66	0.53
	48 h Fermentation parameter														
pH	6.67	6.66	6.62	6.60	6.61	6.61	6.61	6.65	6.66	6.64	6.64	<0.01	0.33	0.96	0.02
NH_3_-N ^1^ (mg/100 mL)	14.59	15.33	14.12	14.39	14.39	14.26	14.97	14.56	14.56	15.72	15.68	0.17	0.31	0.09	0.03
TVFA ^2^ (mmol/L)	78.67	59.34	67.13	62.49	59.01	67.81	57.83	71.45	91.58	55.20	60.98	3.25	0.90	0.82	0.96
Acetate (%)	60.44	64.00	65.05	65.47	61.98	66.64	63.99	65.92	72.50	65.43	68.31	0.96	0.24	0.04	0.89
Propionate (%)	17.42	17.22	17.02	16.64	17.22	16.58	17.16	16.73	14.21	17.30	16.19	0.27	0.55	0.14	0.93
Isobutyrate (%)	2.43	2.02	1.91	1.94	2.29	1.79	2.10	1.90	1.48	1.92	1.70	0.08	0.41	0.04	0.86
Butyrate (%)	13.10	11.37	11.09	10.82	12.42	10.41	11.49	10.59	8.00	10.67	9.51	0.41	0.40	0.03	0.94
Isovalerate (%)	4.73	3.89	3.62	3.69	4.34	3.31	3.81	3.51	2.80	3.41	3.11	0.16	0.50	0.03	0.77
Valerate (%)	1.89	1.49	1.37	1.43	1.75	1.28	1.45	1.35	1.01	1.27	1.17	0.07	0.31	0.02	0.80
A/P ^3^	3.47	3.72	3.94	3.95	3.60	4.02	3.73	3.94	6.01	3.78	4.25	0.21	0.52	0.14	0.96

^1^: ammoniacal nitrogen; ^2^: total volatile acid; ^3^: acetate: propionate ratio.

**Table 7 microorganisms-12-01825-t007:** Primer sequence and optimized annealing temperature.

Name	Primer	Citation	Annealing Temperature
Archaea 806F/958R	5’-ATT AGA TAC CCS BGT AGT CC-3’ 5’-YCC GGC GTT GAM TCC AAT T-3’	[16,17]	55.0 °C
Bacteria BAC27F/EUB338R	5’-AGA GTT TGA TCC TGG CTC AG-3’ 5’-GCT GCC TCC CGT AGG AGT-3’	[17,18]	55.0 °C
ANME-1 1337F/1724R	5’-AGG TCC TAC GGG ACG CAT-3’ 5’-GGT CAG ACG CCT TCG CT-3’	[19]	54.4 °C
ANME-2c 468F/736R	5’-CGC ACA AGA TAG CAA GGG-3’ 5’-CGT CAG ACC CGT TCT GGT A-3’	[20]	56.3 °C
SRB 691F/826R	5’-CCG TAG ATA TCT GGA GGA ACA TCA G-3’ 5’-ACA TCT AGC ATC CAT CGT TTA CAG C-3’	[21]	60.0 °C

**Table 8 microorganisms-12-01825-t008:** The total number of archaea, bacteria, SRB, and ANME in fermentation broth with different sulfur concentrations (log_10_ no. of copies).

Items	Treatments											SEM	*p*-Value	*p*-Value	
	0.4%	0.6%	0.8%	1.0%	1.2%	1.4%	1.6%	1.8%	2.0%	2.2%	2.4%			L	Q
Archaea (log_10_ no.of copies)	5.09 ^a^	5.62 ^c^	5.52 ^ab^	5.13 ^a^	5.14 ^a^	5.11 ^a^	5.42 ^b^	5.15 ^ab^	5.08 ^a^	5.48 ^ab^	5.84 ^d^	0.08	<0.01	<0.01	<0.01
Bacteria (log_10_ no.of copies)	6.95	6.65	6.80	6.14	6. 59	6.83	6.84	6.90	6.56	6.56	6.59	0.08	0.98	0.48	0.63
ANME-1 ^1^ (log_10_ no.of copies)	2.77 ^cd^	2.82 ^de^	2.19 ^a^	2.70 ^c^	2.44 ^b^	2.49 ^b^	3.10 ^h^	2.73 ^c^	2.69 ^c^	2.96 ^fg^	2.88 ^ef^	0.08	<0.01	<0.01	<0.01
ANME-2c ^2^ (log_10_ no.of copies)	3.41 ^a^	3.72 ^cd^	3.50 ^ab^	3.63 ^bcd^	3.61 ^bcd^	3.59 ^bc^	4.06 ^f^	3.78 ^de^	3.69 ^cd^	3.96 ^f^	3.91 ^ef^	0.06	<0.01	<0.01	0.33
SRB ^3^ (log_10_ no.of copies)	1.56 ^a^	1.60 ^ab^	1.48 ^a^	1.46 ^a^	1.53 ^a^	1.45 ^a^	1.72 ^bc^	1.54 ^a^	1.52 ^a^	1.77 ^c^	1.98 ^d^	0.05	<0.01	<0.01	<0.01

^1^: methane anaerobic oxidation archaea-1; ^2^: methane anaerobic oxidation archaea-2c; ^3^: sulfate-reducing bacteria. Note: The letters a–h denote significant mean differences, as established by Tukey’s honest significant difference (HSD) post-hoc analysis, with a significance level of *p* < 0.05.

## Data Availability

The data supporting this study’s findings are available upon request from the corresponding author.

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
