# Peer review of "The Mechanism of Sodium Sulfate Coupled with Anaerobic Methane Oxidation Mitigating Methane Production in Beef Cattle"

_microorganisms, 2024, doi:10.3390/microorganisms12091825_

Round 1
Reviewer 1 Report
Comments and Suggestions for Authors
Thank you for the invitation to review your work, however I consider that it cannot yet be published in the form presented.
summary
please indicate the treatments you use
if you are going to use abbreviations please indicate their meaning e.g. Na2SO4
introduction
please elaborate on the uses of Na2SO4 and doses used, if there is any work done in vivo.
what is your hypothesis?
in your objective you do not mention that you are going to perform PCR tests or correlations, or are the new methods you are proposing and why?
materials and methods
line 54. what is S ? you do not mention its meaning.
why do you use 0.4 % S as a basis ?
line 50-54. How many treatments are there in total because you mention only 11, but there are 12 consultations with the control group that you mention as a target?
table 1. your data should be expressed in g/kg DM not in %.
You should include the amount of g used in all your S treatments?
Indicate the meaning of all your abbreviations at the end of the table.
line 109. what is the objective for making correlations?
Results
figure 1. your results are missing from your target group.
figure 2. your results are not understood, there are two graphs in the same position.
The objective of your work is to see the methane production with the inclusion of your different treatments but you only present them in one graph, in which the values are not visible. I think you should present the data in ml CH4 / g DM incubated in a table and make the corresponding statistics.
In your results you mention that pH and VFA were measured at 24 and 48 hours, but in your materials and methods you do not mention this.
I think you should make bigger tables or adjust your results because they get lost and mixed up.
In addition, in all your tables and graphs you should increase your data of your target group which indicates if any of your results could mitigate methane production with their use.
conclusions
As you do not present your methane results statistically analysed, you only present a graph which is not clearly visible, I could not affirm that methane production is decreased by using S.
Author Response
Thank you very much for taking the time to review this manuscript. I have made some revisions and added supplementary explanations based on your suggestions, which you can find in the following text. Please see the attachment!

Reviewer 2 Report
Comments and Suggestions for Authors Abstract: The experimental design should be reported. The results should be better explored. Acronyms should be defined before their use. A conclusion and a recommendation should be included. Keywords: do not repeat the words in the title; do not use acronyms. Introduction: A hypothesis is necessary. Material and methods. The number of the ethics committee for studies involving the use of animals should be included A table with the nutritional composition of the ingredients in the experimental diet should be included. Table 1 - include the contents of dry matter, ether extract, total carbohydrates and non-fibrous carbohydrates. How much food was offered to the animals? What was the proportion of leftovers? How many liters of water were offered? Before collecting the ruminal fluid, how long did the animals receive the diet?
Author Response
Thank you very much for taking the time to review this manuscript. I have made some revisions and added supplementary explanations based on your suggestions, which you can find in the following text. Please see the attachment.

Reviewer 3 Report
Comments and Suggestions for Authors
In general an interesting manuscript.
What is the impact of sodium? Pls. discuss.
Line 8: Typo in "Conservation" should be corrected to "Conservation."
Line 9: Missing space between "e-mail:" and the actual email address.
line 12: in vitro-->Italic
Line 13: Typo in "oxidization" should be corrected to "oxidation."
Line 17: "Negatively correlate" should be corrected to "negatively correlated."
line 17: Na2SO4-->use subscripts
line 27: 3-5% of the total global greenhouse gas emissions: Pls. check for other Sources, which state that the CH4 contribution is significantly higher.
Line 48: Typo in "sulfur" should be consistent throughout the document, either as "sulfur" or "sulphur.
line 52: , to 0.6%--> remove "to"
Line 39: "methane reduction ability" should be better worded as "capacity to reduce methane."
Line 56: Missing article before "permanent rumen fistula," should be "with a permanent rumen fistula."
Line 71: "Tianqing indicator" should specify what kind of indicator it is (e.g., "Tianqing pH indicator").
Line 83: "Cumulative gas production was recorded manually" would be clearer if written as "The cumulative gas production was recorded manually."
Line 86: "The pH was measured" would be clearer if written as "pH was measured."
Line 106: Sentence starting with "The General Linear Model (GLM) in SPSS 27.0" needs clarity; consider rephrasing for better readability.
Line 119: The phrase "had the highest gas production" could be rephrased as "showed the highest gas production."
Line 121: "Decreased linearly with the increase in sulfur content" should be clarified, possibly specifying which gas.
Line 128: Missing period after the sentence in the caption of Table 3.
Line 129: Missing punctuation and clarity needed in the heading "The effect of different sulfur concentrations on the gas composition of rumen in vitro fermentation."
line 131: H2S, use subscript
Line 138, Fig. 2: There is no Need to shift the x-axis of the 2nd chart, this creates confusion.
Line 156: The description "without heteroduplex formation" could be clarified for non-expert readers.
Line 169: Missing capitalization in "Annealing temperature" under Table 7.
line 182, 263: in vitro-->Italic
Line 194: Typo in "assay" should be "essay" if not referring to a test.
line 197: remove "research"
Line 208: "Uptick" is too informal; consider using "increase."
Line 257: "Nonetheless, further investigations are necessary" could be rephrased for a stronger conclusion.
Line 275: "Original draft preparation" should be written consistently, possibly as "Writing—original draft preparation."
Line 281: The phrase "Data are available on request" is not good, Can You not provide them in a repository with DOI?
Comments on the Quality of English LanguageLanguage is OK.
Author Response

(The authors gave the same response as above.)

Round 2
Reviewer 1 Report
Comments and Suggestions for Authors
Even their work needs improvement.
introduction
1. More work is needed to justify the use of sodium sulphate in ruminants.
2. I am not very clear about your objective because you only mention that with the use of sulphate you can decrease the methane emotions, but what kind of sulphate, please clarify.
materials and methods
3. I am not clear about your treatments because I don't know if you supplement 0.4% sodium sulphate or not, and if I supplement it can't be your control diet because it has already been modified.
4. also in your introduction you mention that the use of 0.2% can have effects on the rumen microbiota because I do not use this dose?
5. The animals do not eat %, please express your diet in g/ kg dry matter, or in g per day /kg metabolic live weight.
6.line 127 should give the meaning of your equation.
7. indicate in your table the amount of Na2SO4 in your diet, you do not present it and it is your additive.
Results
1. you should check your graphs and tables of results and at the bottom place the meaning of all your abbreviations.
2. your results are too close together and not understandable and difficult to understand, e.g. Table 3.
3. table 4. in which units is methane ?
4. check all your tables and put the units in which your results are expressed.
5. line 156-163 has different letter numbers.
discussion
1. your paragraph 4.1 is very large and difficult to understand.
2. you do not present the same sub-indices as your results because?
conclusion
How many in vitro gas experiments did you perform or are you referring to the levels of Na2SO4 you used?
please define well if you used S or sulphur or Na2SO4, your whole document is very confusing.
Author Response
Thank you very much for taking the time to review this manuscript. I have made some revisions and added supplementary explanations based on your suggestions, which you can find in the following text. Please see the attachment. |
